# Design of a Mechanical Part of an Automated Platform for Oblique Manipulation

**Miroslav Blatnický [1], Ján Dižo [1],\*, Milan Sága [2] , Juraj Gerlici [1] and Erik Kuba [1]**

[1] Department of Transport and Handling Machines, Faculty of Mechanical Engineering, University of Žilina, Univerzitná 8215/1, 01026 Žilina, Slovakia; miroslav.blatnicky@fstroj.uniza.sk (M.B.); juraj.gerlici@fstroj.uniza.sk (J.G.); erik.kuba@fstroj.uniza.sk (E.K.)

[2] Department of Applied Mechanics, Faculty of Mechanical Engineering, University of Žilina, Univerzitná 8215/1, 01026 Žilina, Slovakia; milan.saga@fstroj.uniza.sk

\* Correspondence: jan.dizo@fstroj.uniza.sk; Tel.: +421-41-513-2560

**Abstract:** Handling machines are increasingly being used in all sectors of the industry. Knowledge of the theory of transport and handling machines are basic prerequisites for their further technical development. Development in the field of manipulators is reflected not only in their high technical level, but also in increasing safety and economy. The article presents results of research focused on the complete engineering design of a manipulator, which will serve as a mean of the oblique transport of pelletised goods. The manipulator takes the form of a platform moving between two destinations by means of an electromotor. The engineering design of the platform including the track and a working principle is described. The design includes analytical and numerical calculations of main loaded components of the platform. Extensive functional and dimensional calculations serve as the base for preparation of the technical documentation. An important step will be the creation of a parametric model of the force and moment load acting on a platform drivetrain. Based on this, optimal parameters of an electromotor and its dimensional calculation are performed.

**Keywords:** manipulation; design; analytical calculation; numerical analysis; manipulation platform

## 1. Introduction

Recently, it has become possible to meet handling machines of various forms almost at every turn. There are mainly transport means, robotics, lifts, cranes, trolleys etc. Their use is not only in transport, construction or health care services, but their applications are implemented also in other branches of industry, such as storage. Storage is an inseparable part of material flow in every sphere of the economy. The need for material storage of every kind rises due to different timings of production and consumption, different flow of material in an individual part of a logistic chain at every level. Storage is a necessary part of production technology as well.

The problem of warehouses can be studied from different points of view, e.g., from the building point of view, warehouse organisation, used technology etc. In term of logistic objects, the attention is mainly paid to three kind of warehouses, and these are warehouses of bulk materials, warehouses of palletised good and metallurgical materials [1,2]. When one wants to decide which particular kind of warehouse has to be chosen, it is necessary to draw up a thorough analysis of the problem. All requirements of a new device have to be taken into account, because the use of the state of art of technology may not bring the highest functional and economical effects. Handling and transport operations are connected with a relatively large number of workers, and therefore modern engineering production does not function without precisely solved manipulation [3–6].

The objective of this article is to achieve the described requirements by means of implementation of a design of a platform manipulator (Figure 1), which is intended to be mounted in a warehouse. It will transport palletised material on standardised pallets with dimensions 1.2 × 0.8 m with the maximal weight up to 225 kg (up to 300 kg exceptionally) and up to the maximal height of 5 m.

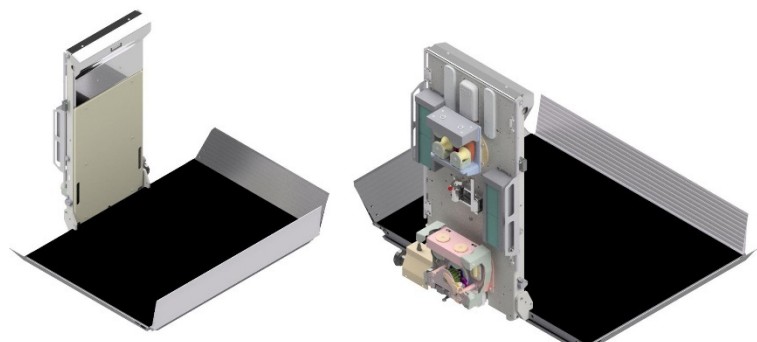

**Figure 1.** A three-dimensional model of the design of the created platform manipulator.

A modern warehouse is not only for material storage. In principle, it is the set of warehouse devices, transport, lifting, sorting and packing machines etc. which can create whole transport lines. The choice of a transport system in a warehouse is a decisive phase of warehouse establishment. The proper choice of the system influences technical, functional, economical and ergonomic aspects of processes during its entire lifetime [1,7].

## 2. Analysis of Theoretical Aspects Influencing the System Choice

Basic data of a warehouse, for which a manipulator has to be designed, are quantified by technical factors and storage possibilities of the given plant. The kind and amount of stored material and warehouse capacity are the main input quantities. In this research, main attention is focused on the infrastructure in term of communications, building layout as well as fire protection. Besides that, we have to take into account existing storage facilities, such as racks, conveyors, further auxiliary means, i.e., pallets, binding and gripping means [8–10], technical means, i.e., electric and electronic elements, compressed air distribution and others. The main criteria, which were taken into account in the process of the proper system choice, are technical and ecological safety, economy, reliability, life cycle, demands on operation, and affordability. When all the described parameters were accepted, the use of a forklift truck seemed to be the easiest form of manipulation of material. However, due to the character of the transported material, it must not be exposed to exhalants produced by a conventional combustion engine regardless of whether they are gasoline, gas [11,12], diesel [13,14] engines or hybrids [15]. Although there are forklift trucks produced with an electric drivetrain [16–18], in consideration of its acquisition costs and its minimal other use, the purchase of an electric forklift truck was refused. Moreover, the parameters of the transported material would lead to a dangerous situation from the dynamic properties point of view.

Thus, if we do not consider warehouses in which pallets are transported by means of forklift trucks, we have to focus on such warehouses in which material is handled by other devices. These devices are generally automatised and controlled by computers [19–21]. However, such a solution suits a warehouse with large capacity and mainly for large assortment, where it is possible to easily reach every one of them. Very narrow aisle trucks, which move in the aisle of a warehouse fully meet that requirement [22,23]. The use of a transelevator [24] was assessed as unprofitable, and therefore it was not considered. An implementation of a shelf stacker was not rejected, because this solution satisfies demands on one hand in terms of the geometrical parameters of the warehouse and on the other hand in terms of the special environment of the warehouse (humidity, temperature and purity) as well as in terms of the price. Figure 2 shows a partial three-dimensional view of the workplace, for which the manipulator is designed. It is obvious, that neither a common elevator cannot be

installed in the warehouse, because its structure would cause as an obstacle to the emergency exit. After consideration and acceptance of all facts described above, the solution of an oblique transport of material by means of the platform was suggested. Such a solution includes all needed properties and meets all operational conditions.

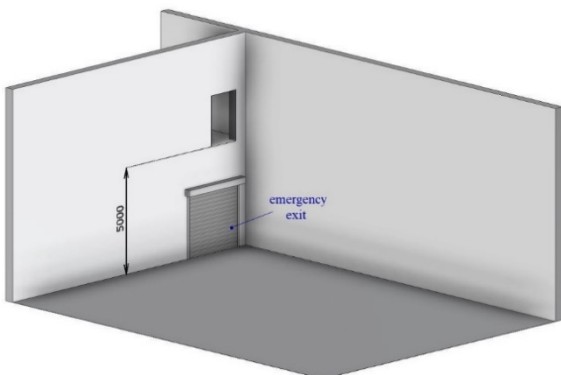

**Figure 2.** A partial three-dimensional view of the workplace for placement of the designed manipulator.

## 3. Design of the Accepted Solution

The design of the handling machine consists of two individual, but interconnected parts.

First, it was necessary to define all external loads acting on the device. Among them, the total weight of the platform and the weight of transported material are the most important input parameters. This mutual load acts on the whole bearing structure and it imposes requirements on parameters of the used drivetrain, i.e., to the drive motor power.

The other part of the design consists of the design of a track, on which the platform will move. It is necessary to take into account its structure, mounting onto a wall, drivetrain mechanism, reliability and safety. The design methodology of such a specific kind of a manipulation machine comes from the STN EN 81-40 standard [25]. As structural designs have to be performed in accordance with the industrial property office requirements, a survey of current technical solutions was carried out. Based on the results, the engineering design of the platform can be used for the intended operational conditions. The platform will move on two steel tubes, which represent the guidance of the platform. The steel tubes with a circular cross-section are mounted on supporting columns. The guidance including columns can be fixed on the wall; the mechanism drivetrain consists of an electromotor and a gearbox, which are built in a platform skeleton. The power transmission is ensured by means of a pinion and a gear rack, which are the part of the guiding track.

The steel structure loaded is during operation by gravitational forces, which cause the strain change. If the distribution of forces, which act on the structure, are improperly analysed, the platform structure would be dimensioned insufficiently and it could result in dangerous operation. The STN EN 81-40 standard [25] defines the minimal loads of platforms of 250 kg·m$^{-2}$. It is necessary to prove that a platform is able to transmit such loads. As the designed device will transport palletised material on the pallet with dimensions of $1.2 \times 0.8$ m, the carrying capacity of the platform must be at least of 240 kg. Therefore, the suggested carrying capacity meets the requirements of the minimum carrying capacity. A conceptual design of the solved manipulator is shown in Figure 3. The solved platform will move in two trolleys placed one above the other and which are rotationally connected with the platform. The trolleys tilt during the platform operation, which results in the change of the forces distribution in the structure. The distance between rotating points of both trolleys is of 0.594 m (Figure 4).

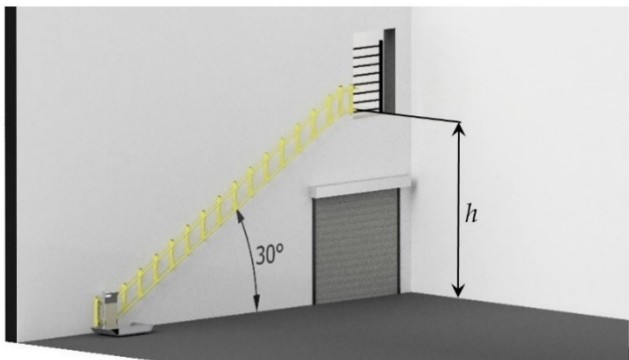

**Figure 3.** A conceptual design of the manipulator.

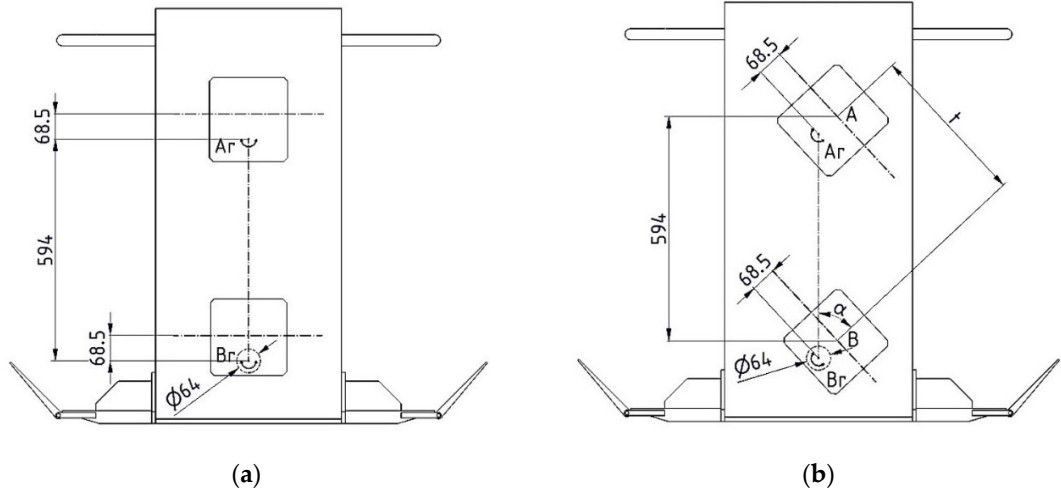

(**a**)  (**b**)

**Figure 4.** The main dimensions of the travel mechanism of the platform: (**a**) when it moves in a straight line; (**b**) when it moves on a slope.

This distance is constant despite the varying track slope. The distance between the two rotating points of the trolleys and the longitudinal axes of both track tubes is constant as well. It is 0.0685 m. Because the points of rotation of both trolleys (Figure 4, points $A_r$ and $B_r$) are on the mutual vector line of the gravitational acceleration, the vertical distance of resulted reaction effects must be 0.594 m. It is necessary to realise, that the track slope change influences the perpendicular distance of individual reaction forces, in Figure 4a marked as $t$.

It is possible to determine parameter $t$ analytically by the following equation:

$$t = 0.594 \cdot \sin \alpha, \tag{1}$$

Where $\alpha$ (°) is the track slope. The constant vertical distance of the trolleys' reactions also influences the calculation of the horizontal reactions. These reactions compensate the moment of the platform gravitational force (Figure 5a—$G_1$ force) as well as the moment of the transported material gravitational force (Figure 5b—$G_2$ force). They act on arms marked as $g_1$ and $g_2$. The free body diagram is shown in Figure 5. From it, we can determine the equations of equilibrium as the following:

$$\sum_i M_{iA} = 0 \rightarrow R_{bz} \cdot 0.594 - G_1 \cdot g_1 - G_2 \cdot g_2 = 0, \tag{2}$$

$$\sum_i F_{iz} = 0 \rightarrow R_{bz} - R_{az} = 0. \tag{3}$$

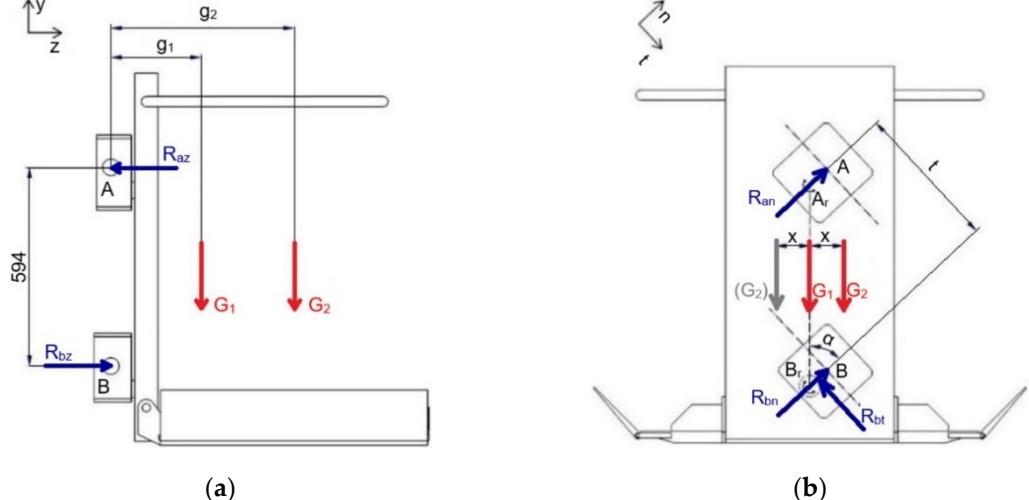

**Figure 5.** Application of the free body diagram method for the platform: (**a**) when it moves in a straight line; (**b**) when it moves on a slope.

Solving Equations (2) and (3) we obtain:

$$R_{bz} = \frac{G_1 \cdot g_1 + G_2 \cdot g_2}{0.594}, \tag{4}$$

and

$$R_{az} = R_{bz}, \tag{5}$$

$$\sum_i F_{it} = 0 \rightarrow G_1 \cdot \sin \alpha + G_2 \cdot \sin \alpha - R_{bt} = 0, \tag{6}$$

$$\sum_i M_{iBr} = 0 \rightarrow R_{an} \cdot t + G_2 \cdot x - R_{bt} \cdot 32 = 0. \tag{7}$$

The solution of Equations (8)–(10) gives

$$R_{bz} = (G_1 + G_2) \cdot \sin \alpha, \tag{8}$$

$$R_{bz} = \frac{R_{bt} \cdot 32 - G_2 \cdot x}{t} \rightarrow R_{an} = \frac{(G_1 + G_2) \cdot \sin \alpha \cdot 32 - G_2 \cdot x}{t}, \tag{9}$$

$$\sum_i F_{in} = 0 \rightarrow R_{an} + R_{bn} - G_1 \cdot \cos \alpha - G_2 \cdot \cos \alpha = 0. \tag{10}$$

From Equation (10) we obtain:

$$R_{bn} = (G_1 + G_2) \cdot \cos \alpha - R_{an} \rightarrow R_{bn} = (G_1 + G_2) \cdot \cos \alpha - \frac{(G_1 + G_2) \cdot \sin \alpha \cdot 32 - G_2 \cdot x}{t}. \tag{11}$$

The validity of the Equations (11) and (12) will be for any value of the angle $\alpha$, if the parameter $x$ (the distance of the gravitational force $G_2$ from the vertical axis between points $A_r$ and $B_r$, Figure 5b) in the right direction will gain a positive value and in the left direction it will gain a negative value. Considering the dimensions of the platform of $1.2 \times 0.8$ m and the centre of gravity location, the parameter $x$ is in the interval from $-0.100$ to $+0.100$ m.

Table 1 contains all geometrical parameters and Table 2 includes all force effects, which are needed for the calculation. The maximal slope (angle $\alpha$) of the track including safety conditions is determined to the value of $\alpha = 47°$.

**Table 1.** Geometrical parameters for calculation.

| Parameter | Value (m) |
|---|---|
| $g_1$ | 0.2730 |
| $g_2$ | 0.5630 |
| $t$ | 0.4344 |
| $x$ | ±0.1000 |

**Table 2.** Determined values of external loads of the platform.

| Considered Weight (kg) | Gravitational Force (N) |
|---|---|
| 154 | 1510.74 |
| 225 | 2207.25 |
| 300 | 2943.00 |
| 375 | 3678.75 |

The engineering design of the platform considers the maximal carrying load capacity of 300 kg, when this load is transported up to the end station. Within the safety conditions, the platform can be loaded up to the weight of 375 kg, but, in such a case, the platform must not move. Otherwise, it could lead to the platform damage. The kerb weight of the platform is 154 kg.

Substituting of values from Table 2 into Equations (1)–(10) gives results of reaction effects introduced in Table 3, where:

$R_{az}$—tangential reaction of wheels of the upper trolley;

$R_{bz}$—tangential reaction of wheels of the bottom trolley;

$R_{an}$—normal reaction of wheels of the upper trolley;

$R_{bn}$— normal reaction of wheels of the bottom trolley;

$R_{bt}$—slope resistance.

**Table 3.** Calculated values of reaction effects considering the geometry of the device and values of loads.

| X = +0.1 m | 225 kg | 300 kg | 375 kg | X = −0.1 m | 225 kg | 300 kg | 375 kg |
|---|---|---|---|---|---|---|---|
| $R_{az}$ (N) | 2786.39 | 3483.74 | 4171.09 | $R_{az}$ (N) | 2786.39 | 3483.74 | 4181.09 |
| $R_{bz}$ (N) | 2786.39 | 3483.74 | 4181.09 | $R_{bz}$ (N) | 2786.39 | 3483.74 | 4181.09 |
| $R_{bt}$ (N) | 2719.17 | 3257.26 | 3795.35 | $R_{bt}$ (N) | 2719.17 | 3257.26 | 3795.35 |
| $R_{an}$ (N) | −307.81 | −437.54 | −567.27 | $R_{an}$ (N) | 708.42 | 917.43 | 1126.44 |
| $R_{bn}$ (N) | 2843.47 | 3474.98 | 4106.5 | $R_{bn}$ (N) | 1827.24 | 2120.01 | 2412.78 |
| $\sum |R_{an}| + |R_{bn}|$ (N) | 3151.28 | 3912.52 | 4673.77 | $\sum R_{an} + R_{bn}$ (N) | 2535.66 | 3037.44 | 3539.22 |

Based on the performed calculation, the greatest load of the platform is in the moment, when the force $G_2$ is shifted by the value of $x$ = +0.1 m. Just this position of the load is chosen for the calculation of resistance forces. The change of the reaction forces due to the platform geometry is calculated by means of the Matlab 2017a program (MathWorks Inc., Natick, MA, USA) [26–28] and their graphs are shown in Figure 6, where a blue line represents the waveform of the reaction force $R_{an}$, a green line depicts the reaction force $R_{bn}$ and a red curve $R_{sum}$ is the waveform of the sum of reaction forces.

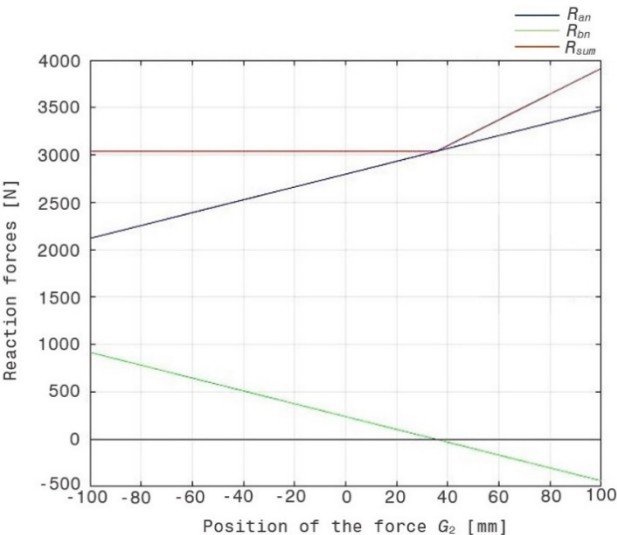

**Figure 6.** Dependence of values of the reaction forces $R_{an}$, $R_{bn}$ and $R_{sum}$ on the platform geometry (position of the centre of gravity $G_2$).

Furthermore, these results are used for the calculation of resistance forces, which arise during the platform operation on the track. Because of the rolling of guiding wheels of the trolleys (Figure 7), rolling resistance arises. This rolling resistance occurs between wheels and pins. Movement of the platform causes elastic deformation of the track as well as the platform wheels. This elastic deformation results in resistance moment arising. As we can see in Figure 7, the upper trolley transmits the reactions $R_{az}$ and $R_{an}$. Four guiding wheels are mounted on the trolley, however they do not ensure the transmitting of the force $R_{az}$, therefore, it is assumed, that a sliding force between the guiding wheels and the spacer rings side surfaces will arise. On the contrary, the force $R_{an}$ causes rolling resistance and pin friction during the operation. The total resistance of the guiding wheel is given by the following formulations:

$$M_{a1} = R_{an} \cdot (e_{a1} + r_{ca1} \cdot f_{ca1}), \tag{12}$$

$$M_{a1} = \frac{(G_1 + G_2) \cdot \sin \alpha \cdot 32 - G_2 \cdot x}{t} \cdot (e_{a1} + r_{ca1} \cdot f_{ca1}). \tag{13}$$

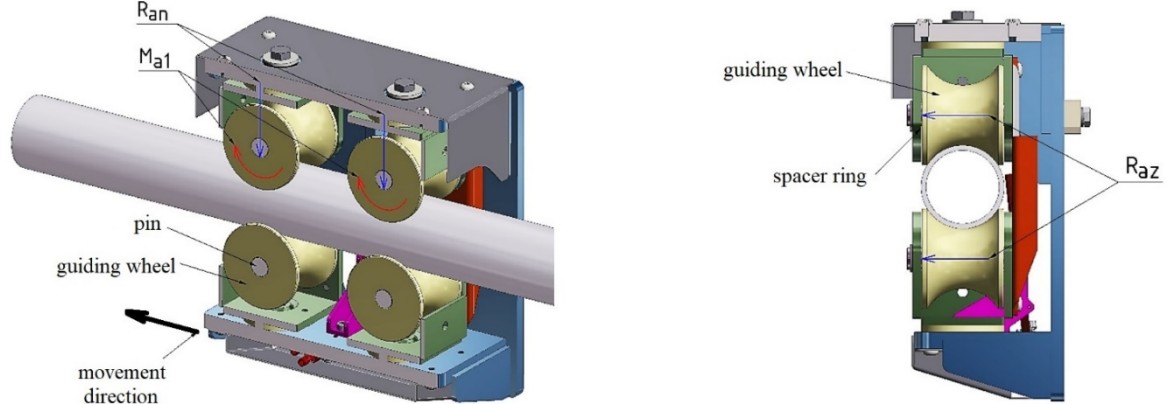

**Figure 7.** Transmission of reaction forces on the upper trolley of the platform.

The friction resistance moment of the spacer ring is calculated as follows:

$$M_{a2} = f_k \cdot \int_r^R x \cdot dR_{az}. \tag{14}$$

The area (Figure 8), on which the force $R_{az}$ acts, is determined:

$$S = \pi \cdot \left(R^2 - r^2\right). \tag{15}$$

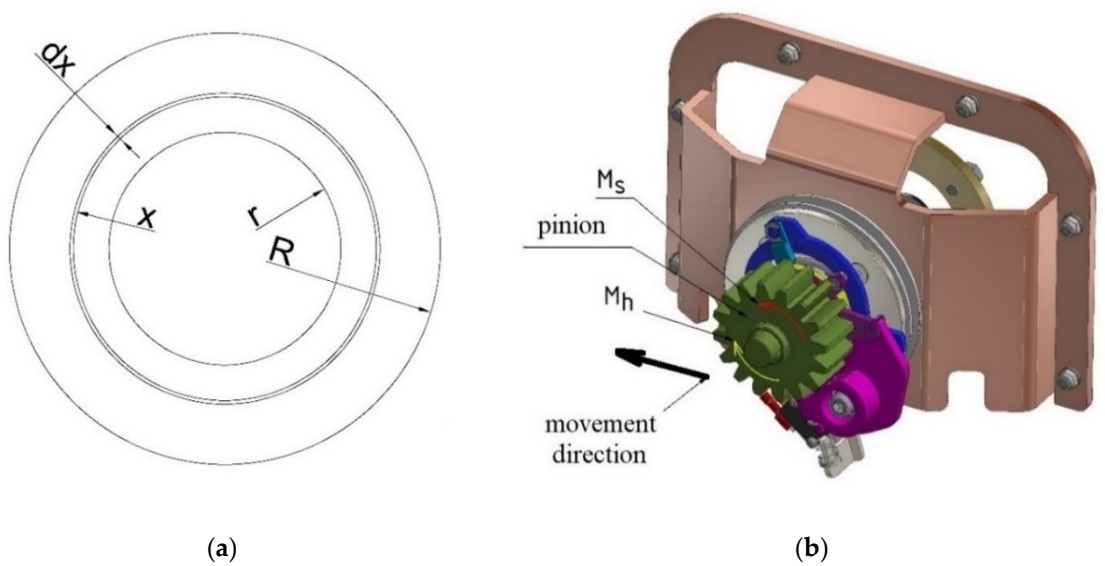

**(a)**          **(b)**

**Figure 8.** (**a**) The elementary area and integration boundaries for determining the resistance moment; (**b**) the reduction in the slope resistance to the output shaft of the platform drivetrain.

The value of the pressure acting on the spacer ring is calculated:

$$p = \frac{R_{az}}{S} = \frac{R_{az}}{\pi \cdot (R^2 - r^2)}. \tag{16}$$

Substituting the Equations (20) and (21) into the Equation (18) we obtain:

$$M_{a2} = f_k \cdot \int_r^R x \cdot \frac{R_{az}}{\pi \cdot (R^2 - r^2)} \cdot dA, \tag{17}$$

where the elementary surface $dA$ is shown in Figure 8 and it is given by the formulation:

$$dA = 2 \cdot \pi \cdot x \cdot dxM. \tag{18}$$

By application of the Equation (16) into the Equation (14) we obtain:

$$M_{a2} = f_k \cdot \int_r^R x \cdot \frac{R_{az}}{\pi \cdot (R^2 - r^2)} \cdot 2 \cdot \pi \cdot x \cdot dx, \tag{19}$$

$$M_{a2} = f_k \cdot \int_r^R x^2 \cdot \frac{2 \cdot R_{az}}{(R^2 - r^2)} \cdot dx = f_k \cdot \frac{2 \cdot R_{az}}{(R^2 - r^2)} \cdot \left[\frac{x^3}{3}\right]_r^R, \tag{20}$$

$$M_{a2} = f_k \cdot \frac{2 \cdot R_{az}}{(R^2 - r^2)} \cdot \frac{R^3 - r^3}{3}. \tag{21}$$

As we can see in Figure 9, the bottom trolley transmits the force reactions $R_{bn}$, $R_{bz}$ and $R_{bt}$. The first reaction $R_{bn}$ is transmitted by means of guiding wheels, which are mounted on rolling bearings.

The force $R_{bz}$ is transmitted to the platform mechanism by means of side rollers, which are mounted on steel pins. The last and at the same time the greatest $R_{bt}$ value is caused due to the slope resistance. The total resistance of the upper trolley wheels is given:

$$M_{b1} = R_{bn} \cdot e_{b1} + 4 \cdot M_{BR}. \tag{22}$$

where $M_{BR}$ (N·m) is the friction moment in a bearing and its value is calculated by means of SKF Bearing Calculator [29]. Substituting appropriate relations into the Equation (17) we obtain the following relation:

$$M_{b1} = \left[ (G_1 + G_2) \cdot \left( \cos\alpha - \sin\alpha \cdot \frac{32}{t} \right) + G_2 \cdot \frac{x}{t} \right] \cdot e_{b1} + 4 \cdot M_{BR}. \tag{23}$$

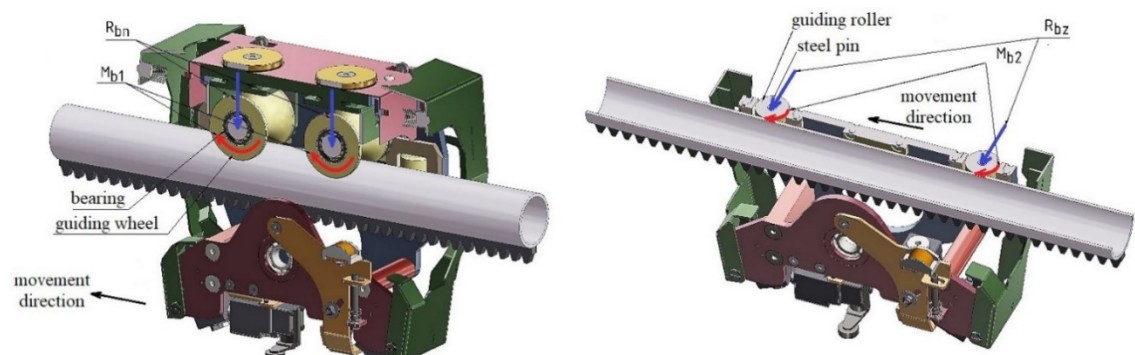

**Figure 9.** Transmission of reaction forces on the bottom trolley of the platform.

The total resistance of the rollers of the bottom trolley is:

$$M_{b2} = R_{bz} \cdot (e_{b2} + r_{cb2} \cdot f_{cb2}), \tag{24}$$

$$M_{b2} = \frac{G_1 \cdot g_1 + G_2 \cdot g_2}{594} \cdot (e_{b2} + r_{cb2} \cdot f_{cb2}). \tag{25}$$

and the slope resistance is:

$$M_S = \frac{d}{2} \cdot R_{bt}, \tag{26}$$

$$M_S = \frac{d}{2} \cdot (G_1 + G_2) \cdot \sin\alpha. \tag{27}$$

where $d$ (mm) is the thread effective diameter of the pinion (Figure 8b). All resistance effects are reduced on this diameter.

Using Equations (16)–(33) and values listed in Tables 3 and 4, we obtain:

$$M_{a1} = R_{an} \cdot (e_{a1} + r_{ra1} \cdot f_{ra1}) = 437.54 \cdot (0.00004 + 7.5 \cdot 0.00005) = 0.182 \, \text{N} \cdot \text{m}, \tag{28}$$

$$M_{a2} = f_k \cdot \frac{2 \cdot R_{az}}{R^2 - r^2} \cdot \frac{R^3 - r^3}{3} = 0.1 \cdot \frac{2 \cdot 3483.74}{13^2 - 7^2} \cdot \frac{(0.013)^3 - (0.007)^3}{3} = 3.588 \, \text{N} \cdot \text{m}, \tag{29}$$

$$M_{b1} = R_{bn} \cdot e_{b1} + 4 \cdot M_{BR} = 3474.98 \cdot 0.00004 + 4 \cdot 69.7 = 0.418 \, \text{N} \cdot \text{m}, \tag{30}$$

$$M_{b2} = R_{bz} \cdot (e_{b1} + r_{rb2} \cdot f_{rb2}) = 3483.74 \cdot (0.00003 + 0.004 \cdot 0.05) = 0.800 \, \text{N} \cdot \text{m}, \tag{31}$$

$$M_s = \frac{d}{2} \cdot R_{bt} = \frac{0.064}{2} \cdot 3257.26 = 104.23 \, \text{N} \cdot \text{m}, \tag{32}$$

$$\sum_i M_{oi} = M_{a1} + M_{a2} + M_{b1} + M_{b2} + M_s =$$
$$= 0.182 + 3.588 + 0.418 + 0.800 + 104.23 = 109.22 \; \text{N} \cdot \text{m} \tag{33}$$

**Table 4.** Input parameters for calculation of resistance forces and moments.

| Parameter | Value | Parameter | Value |
|---|---|---|---|
| $f_{p1}$ | 0.05 | $n_m$ | 46.666 s$^{-1}$ |
| $f_{p2}$ | 0.05 | $g$ | 9.81 m·s$^{-2}$ |
| $r_{p1}$ | 0.0075 m | $\alpha$ | max. 47° |
| $r_{p2}$ | 0.004 m | $v$ | 0.1 m·s$^{-1}$ |
| $e_{a1}$ | 0.00004 m | $i$ | 100 |
| $e_{b1}$ | 0.00004 m | $m$ | 0.004 m |
| $e_{b2}$ | 0.00003 m | $z$ | 16 |
| $\eta_1$ | 0.65 | $M_{BR}$ | 69.7 N·m |
| $\eta_2$ | 0.98 | $R$ | 0.013 m |
| $\eta_3$ | 0.99 | $r$ | 0.007 m |
| $m_1$ | 154 kg | $d$ | 0.064 m |
| $m_2$ | 300 kg | $\omega$ | 2.932 rad·s$^{-1}$ |

The electromotor of the platform, which serves as a drive source, must evolve the torque $M_h$ (N·m) to be greater than the sum of all resistance torques (Equation (34)). We consider the angular velocity of the drive shaft $\omega = 2.932$ rad·s$^{-1}$, which ensures required translational velocity of the platform (Table 4). Hence, the torque of the electromotor on the output shaft is given by the following formulation:

$$P_v = M_h \cdot \omega = 109.22 \cdot 2.932 = 320.23 \; W. \tag{34}$$

Considering the total drivetrain efficiency $\eta_c$, the total power of the electromotor is:

$$P_m = \frac{P_v}{\eta_c} = \frac{320.23}{0.98 \cdot 0.65 \cdot 0.99} = 507.79 \; W. \tag{35}$$

The input parameters for Equations (12)–(27) are listed in Table 4, where:

$f_{p1}$—coefficient of friction of the upper trolley wheels;
$f_{p2}$—coefficient of friction of the bottom trolley wheels;
$r_{p1}$—radius of the pinion of the upper guidance;
$r_{p2}$—radius of the pinion of the bottom guidance;
$e_{a1}$—forward moving distance of the upper guidance wheels;
$e_{b1}$—forward moving distance of the bottom guidance wheels;
$e_{b2}$—forward moving distance of the side guiding rollers of the bottom guidance;
$\eta_1$—gearbox efficiency;
$\eta_2$—bearing efficiency;
$\eta_3$—efficiency of the gear rack and the pinion;
$m_1$—weight of the platform;
$m_2$—weight of the load;
$n_m$—nominal rpm under the load;
$g$—gravitational acceleration;
$\alpha$—slope angle;
$v$—operational speed of the platform;
$i$—gear ratio;
$m$—module of the gear rack and the pinion;
$z$—number of pinion teeth;

$M_{BR}$—friction moment of the bearing;
$R$—outer radius of the annulus area of the bearing;
$r$—inner radius of the annulus area of the bearing;
$d$—reference diameter of the pinion;
$\omega$—angular velocity of the pinion.

Table 5 contains the moments of resistance forces and the corresponding powers of the platform electromotor, which are determined for all calculated loads. The designations mean the following:

$M_{a1}$—friction moment of the upper trolley wheels;
$M_{a2}$—friction moment of an axial pin of the upper trolley;
$M_{b1}$—friction moment of the bottom trolley wheels;
$M_{b2}$—friction moment of the side guiding rollers of the bottom platform guidance;
$M_S$—resistance moment due to the track inclination;
$P_v$—electromotor power without mechanical losses;
$P_m$—electromotor power including mechanical losses.

**Table 5.** Calculated moments for individual loads of the platform and corresponding powers.

| Parameter | 225 kg | 300 kg | 375 kg |
|---|---|---|---|
| $M_{a1}$ (N·m) | 0.128 | 0.182 | 0.235 |
| $M_{a2}$ (N·m) | 2.870 | 3.588 | 4.307 |
| $M_{b1}$ (N·m) | 0.352 | 0.418 | 0.485 |
| $M_{b2}$ (N·m) | 0.641 | 0.801 | 0.962 |
| $M_s$ (N·m) | 87.013 | 104.232 | 121.451 |
| $\sum_i M_{oi}$ (N·m) | 91 | 109 | 127 |
| $P_v$ (W) | 267 | 320 | 374 |
| $P_m$ (W) | 423 | 508 | 593 |

Using the described calculation, we detected that the electromotor with the power of $P_m$ = 500 W and with the torque overload coefficient of $\xi$ = 3.3 is the sufficient solution for the created platform. The dependence of resistance moments and the electromotor power on the load is shown in Figure 10. It indicates values of the electromotor power needed for overcoming resistance moments resulting from the load.

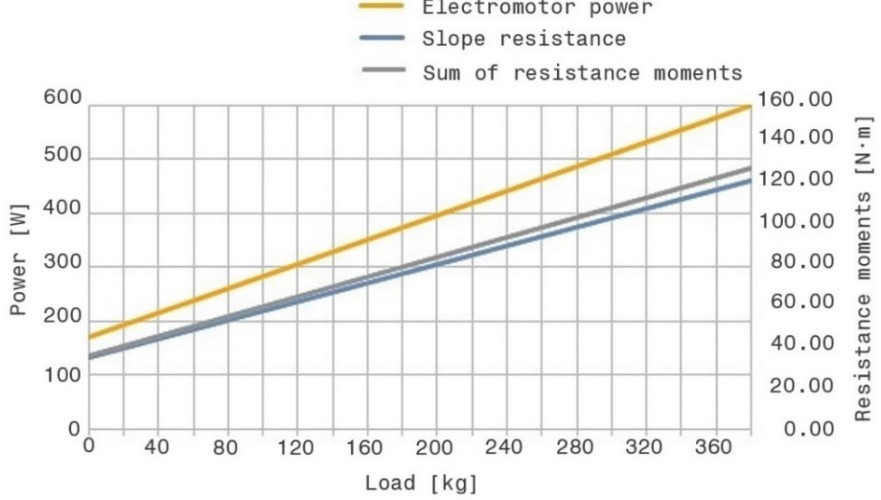

**Figure 10.** Dependence of resistance moments and the electromotor power on the load.

Figure 11 depicts the dependence of the electromotor power on the slope angle at constant loads, namely at 225 kg, 300 kg and 375 kg. We can see that the proposed electromotor is a suitable drivetrain for the designed platform (Figure 1) for the real slope angle of $\alpha = 30°$ (Figure 3).

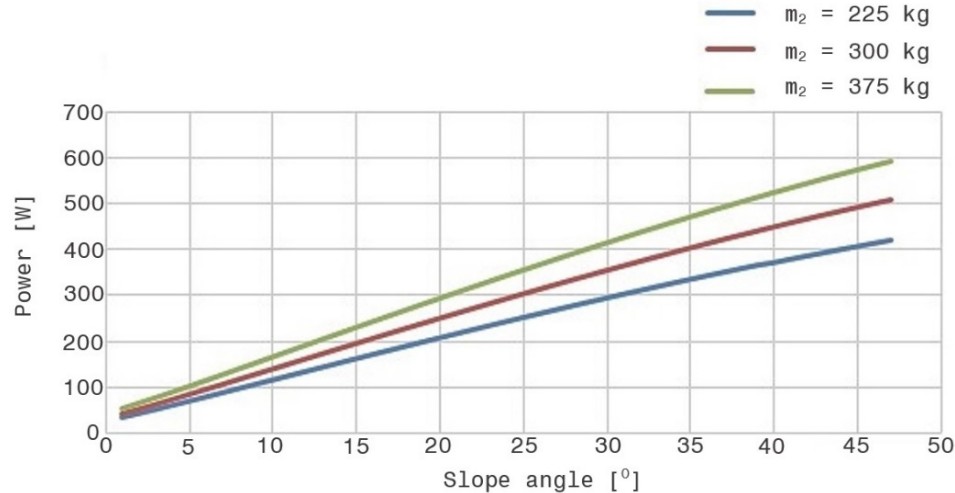

**Figure 11.** The dependence of the track slope angle on the electromotor power under individual platform loads $m_2$.

## 4. Determination of the Most Loaded Component of the Platform

The output shaft of the gearbox is the most loaded component. The drive-shaft of a drive-train mechanism is often the most loaded part, which is imposed not only by mechanical tension resulting from the loads, but also by dynamic loads and vibration occurring during its operation [30,31]. The calculation of the electromotor has proven that the electromotor can be used even for greater loads, but it is necessary to analyse the output shaft (Figures 8b and 12).

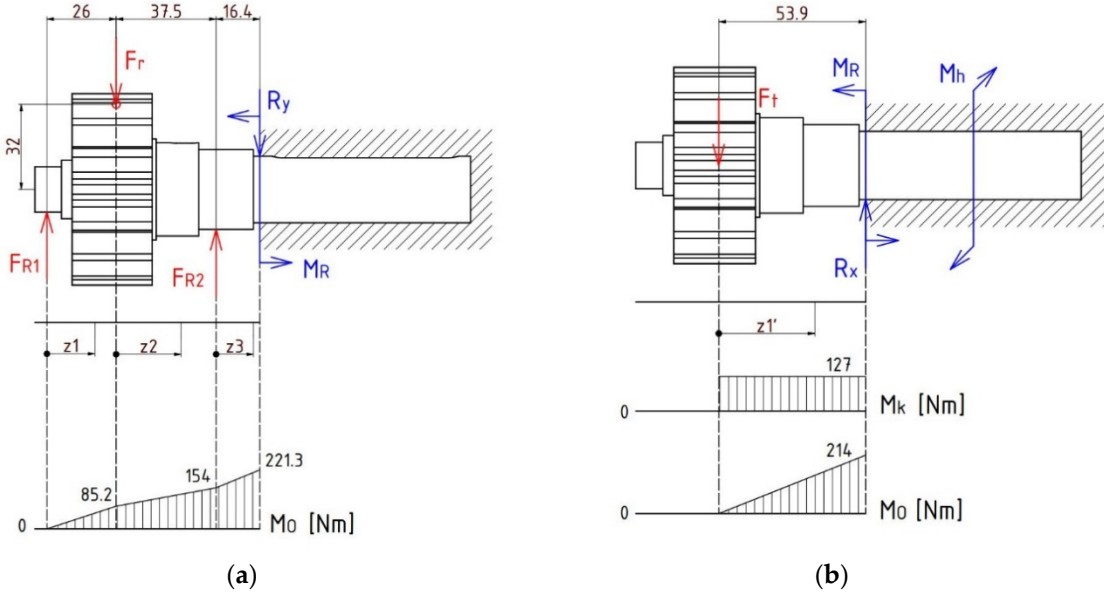

**(a)**                                           **(b)**

**Figure 12.** Scheme and distribution of the load of the output shaft: (**a**) *z-y* plane; (**b**) *z-x* plane.

Individual loading forces are given by the following formulations:

$$F_t = \frac{M_o}{\frac{d}{2}},$$ (36)

$$F_r = F_t \cdot \tan 20°, \tag{37}$$

$$\sum_i M_{iFR1} = 0 \rightarrow F_{R2} \cdot 63.5 - (R_{bn} + F_r) \cdot 26 = 0, \tag{38}$$

$$\sum_i F_{iy} = 0 \rightarrow F_{R1} + F_{R2} - R_{bn} - F_r = 0, \tag{39}$$

$$\sum_i M_{iORy} = 0 \rightarrow M_R - F_{R2} \cdot 16.4 - F_{R1} \cdot 79.9 + F_r \cdot 53.9 = 0, \tag{40}$$

$$M_{O1} = F_{R1} \cdot z_1, \tag{41}$$

$$M_{O2} = F_{R1} \cdot (26 + z_2) - F_r \cdot z_2, \tag{42}$$

$$M_{O3} = F_{R1} \cdot (63.5 + z_3) - F_r \cdot (37.5 + z_2) + F_{R2} \cdot z_3. \tag{43}$$

We can assume that the maximal value of stress will be in the considered coupling, because the maximal value of the bending moment is at the shaft constraint end (diameter is $d_1$ = 25 mm) and other diameters of the shaft are bigger as well as the bending moments in these locations being less. The corresponding bending modulus of section is:

$$W_O = \frac{\pi \cdot d_1^2}{32}. \tag{44}$$

The maximal stress in the *z-y* plane is given as the following:

$$\sigma_{O1} = \frac{M_o}{W_o}. \tag{45}$$

The individual forces depicted in Figure 12 are the following: forces $F_{R1}$ and $F_{R2}$ are forces in rolling bearings, which transmit loads from the platform to a bottom trolley. The end of the shaft is mounted rigidly in a gearbox, which causes reaction forces $R_x$, $R_y$ and reaction torque $M_R$. The drive force is decomposed in the contact to a radial component of a drive force $F_r$ and a tangential component. Forces depicted in Figure 12 always act in the radial direction regarding the shaft axis. Furthermore, Figure 12 shows the tangential component of the drive force $F_t$, the drive torque $M_h$ (in the scheme, it is considered as the reaction torque due to the $F_t$ force), the reaction force $R_x$ and the reaction torque $M_R$. The torque $M_h$ is in the equilibrium with the torque due to the force $F_t$ on the radius of the pitch circle of the pinion. Thus, the force $F_t$ loads the shaft by torsion and by bending. Figure 12 represents the static strength analysis in two planes, namely in the *y-z* plane (Figure 12a) and in the *x-z* plane (Figure 12b).

Substituting particular values into Equations (36)–(45) we obtain the maximal value of the bending stress of $\sigma_{o1}$ = 144.3 MPa. Similar equations are also valid for Figure 12b. The analytical calculation gives the maximal value of the bending stress of $\sigma_{o2}$ = 139.5 Mpa and the maximal value of the torsion stress of $\tau_k$ = 41.4 Mpa. The resulting bending stress in two perpendicular planes is determined by the following equation:

$$\sigma_O = \sqrt{\sigma_{o1}^2 + \sigma_{o2}^2}. \tag{46}$$

When we substitute values to the Equation (45), the resulting value of the bending stress is $\sigma_o$ = 200.7 Mpa. The calculation of the reduced stress according to the von-Misses theory is:

$$\sigma_{RED} = \sqrt{\sigma_o^2 + 3 \cdot \tau^2} = \sqrt{(200.7)^2 + 3 \cdot (41.4)^2} = 213.3 \text{ MPa.} \tag{47}$$

Numerical calculations of the reduced stress in the analysed shaft were performed using the Inventor Nastran software package (Autodesk Inc., San Rafael, CA, USA), which is based on the finite element method [32–35]. The results are shown in Figures 13 and 14. Both analyses reached very similar results. The material of the shaft with the pinion is steel marked 42CrMo4. The yield strength of of the material

is of $R_e = 650$ Mpa. It means that the reached shaft safety coefficient is $k = 2.84$. It is a sufficient value even for the maximal load of the platform of 375 kg.

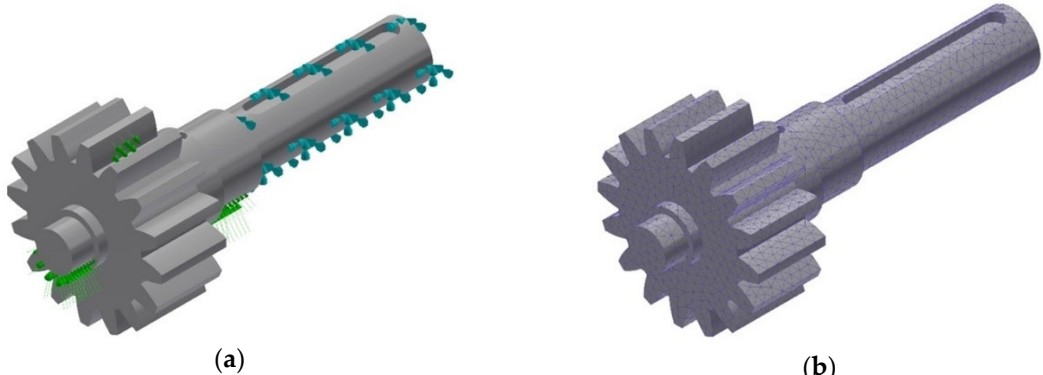

| (**a**) | (**b**) |

**Figure 13.** (**a**) Definition of the boundary conditions for calculation; (**b**) the shaft mesh model.

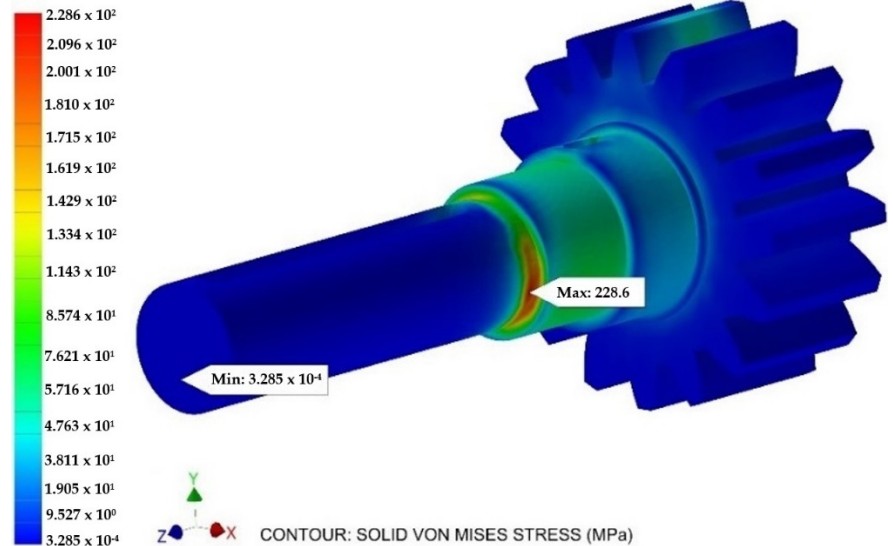

**Figure 14.** Distribution of the stress in the analysed shaft for the maximal considered load.

## 5. A Mathematical Model of a Platform Motion

This section contains the mathematical model of the platform motion during its actuation. The aim of this model is to analyse the waveform of the main acting moments in the mechanism. The mathematical model is described by an equation of motion derived by means of Lagrange's equations of the second kind, which the general form is known as the following:

$$\frac{d}{dt}\frac{\partial E_k}{\partial \dot{q}_j} - \frac{\partial E_k}{\partial q_j} + \frac{\partial E_D}{\partial \dot{q}_j} + \frac{\partial E_P}{\partial q_j} = Q_j, j = 1, 2, \ldots, n, \tag{48}$$

where $E_k$ is the kinetic energy of a mechanical system, $E_D$ is the Rayleigh dissipative function, $E_P$ is the potential energy of a mechanical system, $Q_j$ is the vector of external loads and $q_j$ are generalised coordinates of the mechanical system.

We apply this method for our solved mechanism based on the scheme in Figure 15. We can consider the simplified mechanical model of the platform, which consists of rigid bodies. As is identified in the previous chapter, the drive shaft of the gearbox is the most loaded component. Figure 15 depicted the entire platform mechanism as well as the detail of the driving system.

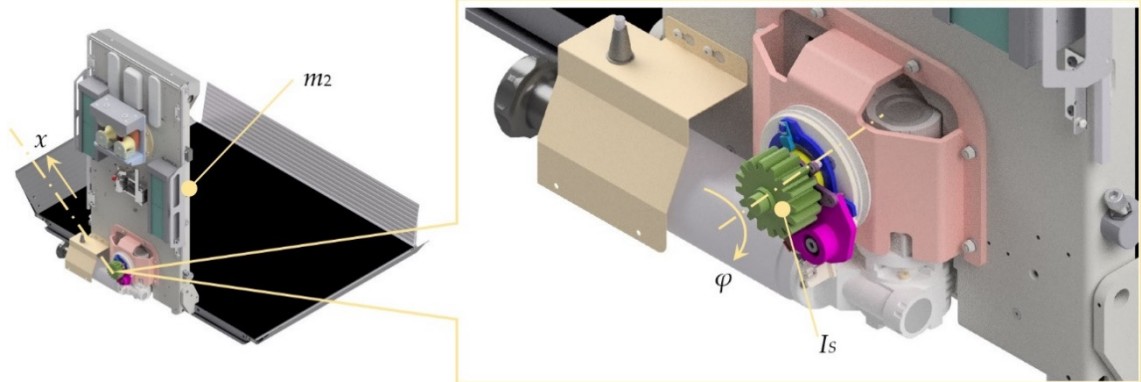

**Figure 15.** Indication of parameters for derivation of the equation of motion of the platform motion.

We consider that the simplified mechanical system has one degree of freedom, i.e., the rotation of the drive-shaft marked as $\varphi$. Moreover, we can consider, that other bodies of the platform perform translation movement in the *x*-axis direction. We need to substitute into Lagrange's equation of the second kind their first time derivation, which represent velocities as the following:

- The rotational velocity of the drive-shaft is $\dot{\varphi}$;
- The translational velocity of other components is $\dot{x}$.

Other rotating components of the platform have a negligible effect on the dynamic properties of the platform in terms of the investigated problem, and therefore their moments of inertia are much smaller in comparison with the moment of inertia of the drive-shaft.

Based on the described assumptions, the mechanical system of the platform is described by means of one equation of motion. For its derivation, we need to identify the individual energies of the mechanical system (Equation (48)), which are as follows:

- The kinetic energy $E_k$:

$$E_k = \frac{1}{2} \cdot I_S \cdot \dot{\varphi}^2 + \frac{1}{2} \cdot m_2 \cdot \dot{x}^2, \tag{49}$$

where $I_S$ is the moment of inertia of the drive-shaft and $m_2$ is total weight of the platform.

As we consider that the motion of the mechanical system of the platform will be described only by the one coordinate, namely the *x* coordinate, we must express the angle $\varphi$ by a constraint equation. The platform is driven by the gearing of the drive-shaft without slipping. Thus, the translational velocity of the platform $\dot{x}$ equals the peripheral velocity $\dot{\varphi}$ of the drive-shaft on a pitch circle with the radius $R_G$. The same relation is valid between translation motion of the platform *x* and angular deviation $\varphi$ of the drive-shaft. Then, the constraint equation is:

$$x = R_G \cdot \varphi. \tag{50}$$

The final form of the kinetic energy is:

$$E_k = \frac{1}{2} \cdot \left( I_S + m_2 \cdot R_G^2 \right) \cdot \dot{\varphi}^2. \tag{51}$$

- The potential energy $E_P$:

$$E_P = m_2 \cdot g \cdot h, \tag{52}$$

where $g$ is the gravitational acceleration and $h$ is the total height which the platform must overcome during operation (Figure 1). Again, we must express the potential energy by means of the $\varphi$ coordinate. When we take into account the inclination angle $\alpha$, we obtain the equation:

$$h = x \cdot \sin \alpha \tag{53}$$

and, considering the Equation (50), we obtain the final form of the potential energy as the following:

$$E_P = m_2 \cdot g \cdot \varphi \cdot R_G \cdot \sin \alpha. \tag{54}$$

- The Rayleigh dissipative function $E_D$:

$$E_D = \frac{1}{2} \cdot b \cdot \dot{\varphi}^2, \tag{55}$$

where $b$ is the coefficient, which includes energy losses in the mechanical system. In this case, it represents mainly friction losses in the mechanism which we can also call passive resistances.

Now, we derivate all energies in accordance with the Equation (41) and we obtain the final form of the equation of motion:

$$\left( m_2 \cdot R^2 + I_S \right) \cdot \ddot{\varphi} + b \cdot \dot{\varphi} = M_h - m_2 \cdot g \cdot R_G \cdot \sin \alpha, \tag{56}$$

where $M_h$ is the drive torque.

Solving of the derived equation of motion of the platform mechanism is performed using the Matlab 2017a program (MathWorks Inc., Natick, MA, USA).

The driving moment is the input to the solution by means of an input file, which contains characteristics of the driving electromotor. For calculation we consider the initial conditions as the following:

- The platform starts from rest,
- For time $t = 0$ s, the initial angular velocity $\dot{\varphi} = 0$ rad $\cdot$ s$^{-1}$, the initial angular deviation $\varphi = 0$ rad,
- Time interval for calculation was set for 0.5 s.

Figures 16–18 show waveforms of the driving torque of the electromotor, the torque of passive resistances and the torque of slope resistance for weight $m_2$ of 225 kg, 300 kg and 375 kg, respectively.

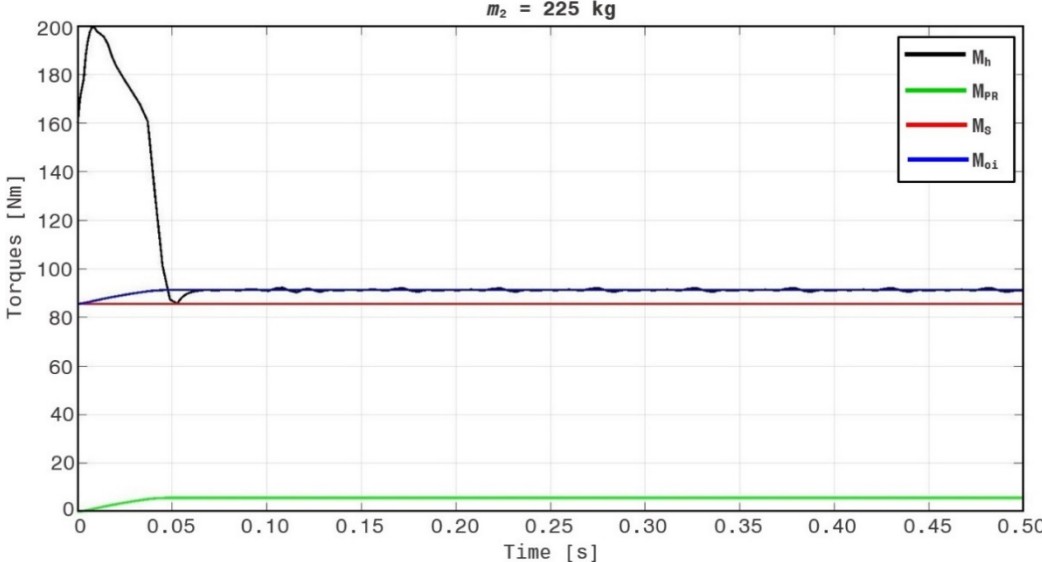

**Figure 16.** Waveforms of investigated torques for the weight $m_2$ = 225 kg.

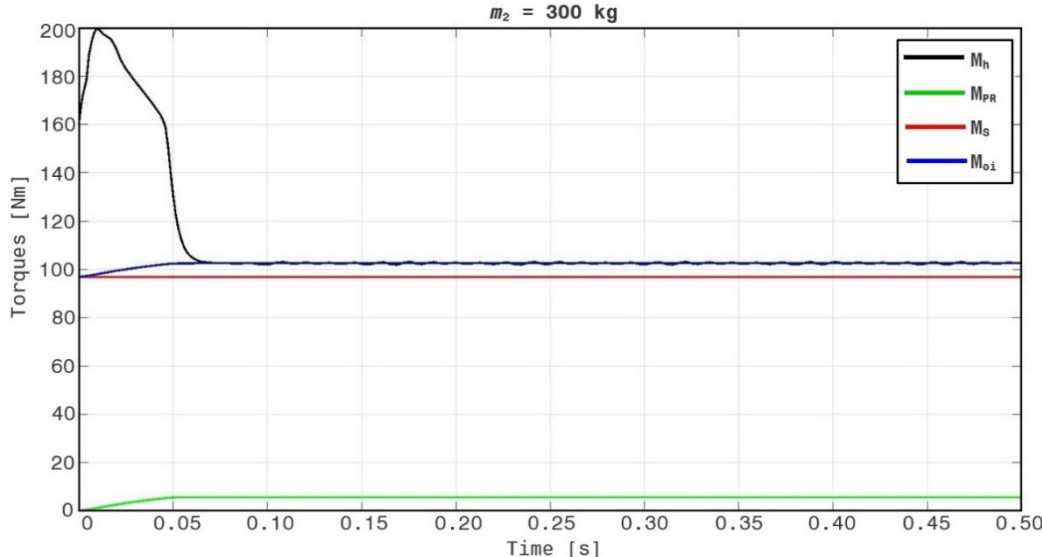

**Figure 17.** Waveforms of investigated torques for the weight $m_2$ = 300 kg.

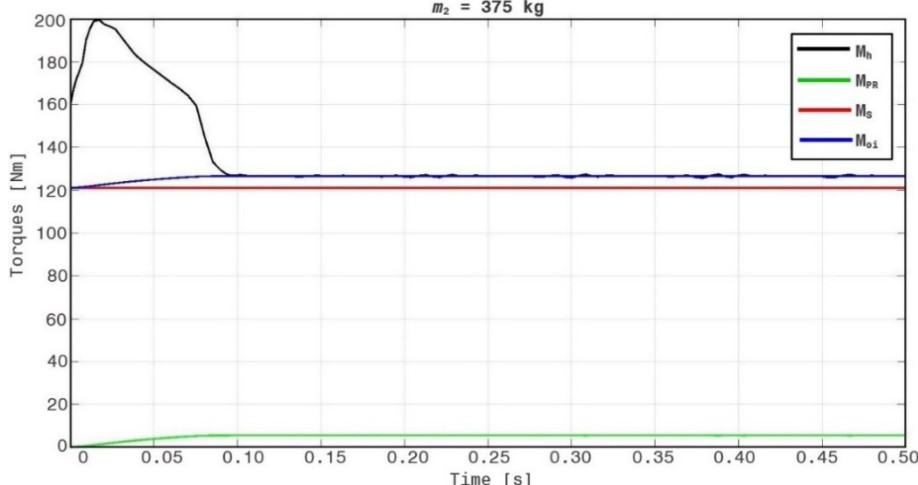

**Figure 18.** Waveforms of investigated torques for the weight $m_2$ = 375 kg.

These figures show that waveforms of the investigated torques correspond to values introduced in Table 5. Individual torques are marked as the following: $M_h$ is the drive torque of the electromotor, $M_{PR}$ is the torque of the passive resistance, $M_S$ is the torque of slope resistance and $M_{oi}$ is the total resistance torque, i.e., $M_{oi} = M_S + M_{PR}$. We can observe that the start of the platform motion is accompanied with increasing drive torque of the electromotor. Then, the drive torque gradually decreases, while the equilibrium of the total resistance torque and the drive torque is reached. Further waveforms depict the steady state of the platform motion.

It is obvious from the figures, that achieving equilibrium of the drive torque and the torque of total resistance depends on the weight. This weight causes directly proportional inertial effects of the mechanism. The change of the weight from 225 kg to 375 kg increases the achieving the steady state approx. twice.

Figure 19 shows acceleration of the platform for its starting. We can see that acceleration of the platform reaches high values. This phenomenon is caused due to the fact that we do not know the exact characteristics of the control system used for controlling the electromotor. Finally, it should be mentioned, that the wavy waveforms of the drive torques in Figures 15–17 and acceleration in Figure 18 are caused due to set errors in the integration method used for calculation of the mathematical model as well as due to dynamic effects of the platform motion.

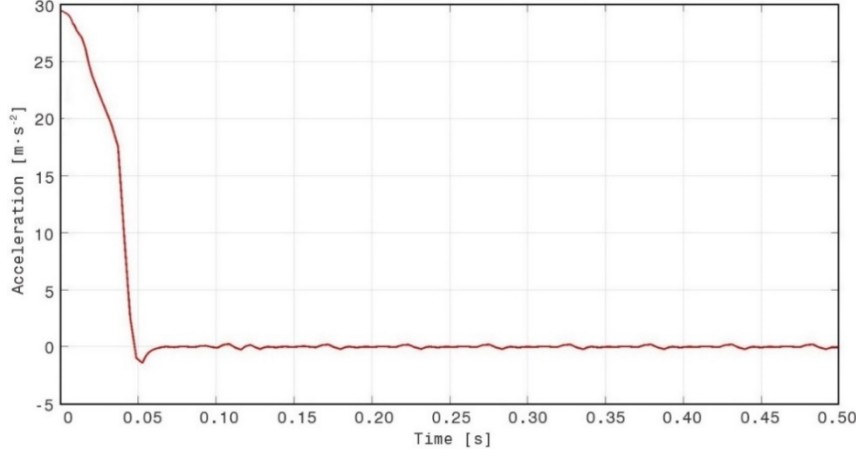

**Figure 19.** Waveform of acceleration of the platform mechanism for time interval of 0.0–0.5 s.

*Eigenfrequencies of the Drive Shaft*

Other dynamic analyses were aimed at the identification of a basic dynamic characteristic of the drive shaft, namely at finding its angular eigenfrequencies $\Omega_i$, or eigenfrequencies $f_i$. The observed eigenfrequency is compared with the angular excitation frequency $\omega$ during operation. Their similar values could lead to undesirable excessive vibrations of the mechanism.

The equation of motion for free oscillation of the drive shaft is:

$$I_S \cdot \ddot{\varphi} + k_S \cdot \varphi = 0, \tag{57}$$

where $I_S$ ($I_S = 0.462$ kg·m$^2$) is the moment of inertia of the shaft and $k_S$ ($k_S = 360{,}430$ N·m·rad$^{-1}$) is its torsion stiffness. This form of the equation of motion (Equation (57)) and basic characteristics of the shaft are valid when we consider the drive shaft as a flexible body defined by its torsion stiffness. When we rewrite Equation (57) to the form:

$$\ddot{\varphi} + \frac{k_S}{I_S} \cdot \varphi = 0, \tag{58}$$

and substitute the known parameters, we obtain the eigenfrequency $f_S = 104.565$ Hz.

In reality, the drive shaft is a flexible body represented as a continuum mass, in which eigenmodes can be investigated by means of, e.g., the finite element method (FEM). Then, the equation of motion for free oscillation will be in the form:

$$\boldsymbol{I}_S \cdot \ddot{\boldsymbol{\varphi}} + \boldsymbol{k}_S \cdot \boldsymbol{\varphi} = \boldsymbol{0}, \tag{59}$$

where $\boldsymbol{I}_S$ is the matric of the moments of inertia, $\boldsymbol{k}_S$ is the stiffness matrix, $\ddot{\boldsymbol{\varphi}}$ is the vector of angular accelerations and $\boldsymbol{\varphi}$ is the vector of angular deviations. After performing the modal analysis in the FEM software, we found out, that the first six eigenvalues are as follows: $f_{S1} = 89.298$ Hz, $f_{S2} = 96.923$ Hz, $f_{S3} = 125.863$ Hz, $f_{S4} = 160.963$ Hz, $f_{S5} = 171.385$ Hz and $f_{S6} = 223.951$ Hz.

When we compare results of eigenfrequencies for the flexible body represented just by its torsion stiffness (Equation (58)) and for the flexible body modelled as a continuum (Equation (59)), we can see that results are slightly different. It is caused because the continuum better represents the real body of the analysed drive shaft. The angular excitation frequency of the drive shaft is $\omega = 2.932$ rad·s$^{-1}$, or $f = 0.466$ Hz. This excitation frequency is much lower than the eigenfrequencies calculated by means of Equation (58) as well as Equation (59). We can conclude that the drive shaft will not reach to the resonance due to the driving in the operation mode.

## 6. Analysis of the Platform Deflection

As the STN EN 81-40 standard allows the maximal deflection ($w_{max}$) of the platform to be $w_{max} = 20$ mm [22], it is necessary to analyse this deflection. In the device operation, such a situation can happen that the total load of the platform will be transmitted to columns either by one tube (upper or bottom) or the load will be distributed to both tubes. For both exceptional cases, dependencies of the load were made (Figures 15 and 16). These dependencies influence the deflection of the end platform point when the distance between columns is changed.

The authors introduce an example of the calculation in the case of transmitting of loads by means of the upper guiding tube.

The load, which is caused by the platform weight and the transported goods, represents only one component of force $A_{yII}$ (the reaction force due to the own weight load of the platform and the goods), because the axis force $F_{oII}$ is considered in the pinion gearing (this is the drive force of the pinion and it acts in the platform motion direction), i.e., it influences the strain of the bottom guiding tube. Figure 20 shows the considered loading forces. The acting force $A_{xII}$ is depicted in the cutting-plane *A-A*.

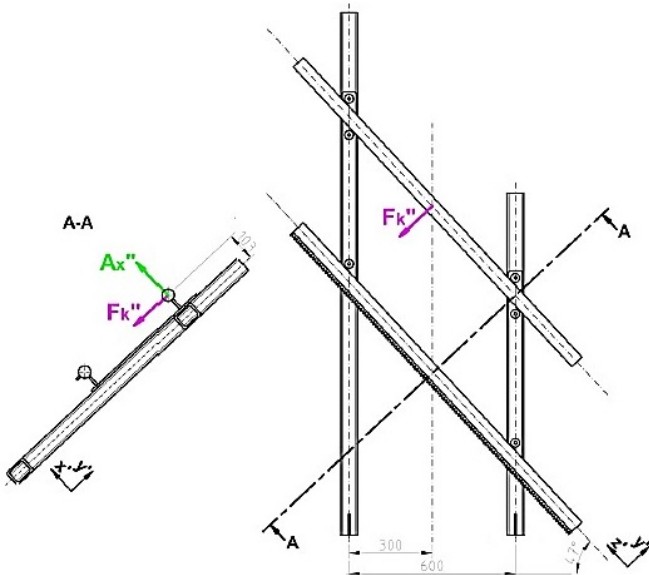

**Figure 20.** Waveform of acceleration of the platform mechanism for time interval of 0.0–0.5 s.

For the relation for calculation of deformation, we can use a simplified model of a beam with two supports representing columns. These supports remove three degrees of freedom. A distribution of the resulting bending moment and the lateral force regarding to the upper guiding tube is shown in Figure 21.

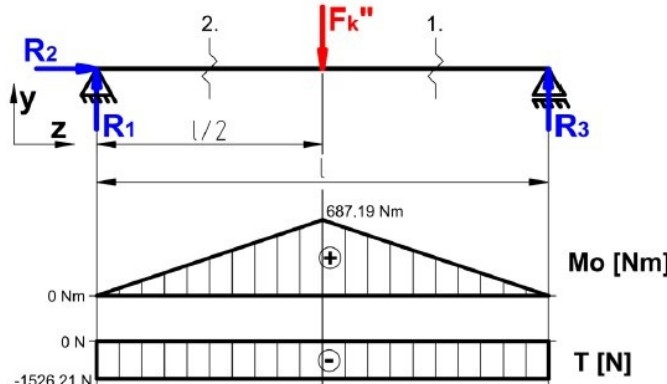

**Figure 21.** A simplified model of a beam and the distribution of the bending moment and the lateral force.

Forces marked as $F_{kII}$ and $A_{xII}$ cause deformation in two different planes. Therefore, we analyse deflections individually (we use the superposition method) and subsequently, we determine the resulting value of the deflection by means of the Pythagorean theorem as follows:

$$\sum_{i=1}^{n} F_{iy} = 0 \Rightarrow R_1 + R_3 - F_k^{II} = 0, \tag{60}$$

$$\sum_{i=1}^{n} F_{iz} = 0 \Rightarrow R_2 = 0, \tag{61}$$

$$\sum_{i=1}^{n} M_{iR_1} = 0 \Rightarrow R_3 \cdot l - F_k^{II} \cdot \frac{l}{2} = 0. \tag{62}$$

We get Equations (63)–(65) for the calculation of reaction force from Equations (60)–(62), respectively:

$$R_3 = \frac{F_k^{II}}{2}, \tag{63}$$

$$R_2 = 0, \tag{64}$$

$$R_1 = F_k^{II} - R_3 = F_k^{II} - \frac{F_k^{II}}{2} = \frac{F_k^{II}}{2}. \tag{65}$$

Calculation of deformation and lateral forces is performed applying the Schwedler's theorem, which comes from Figure 22.

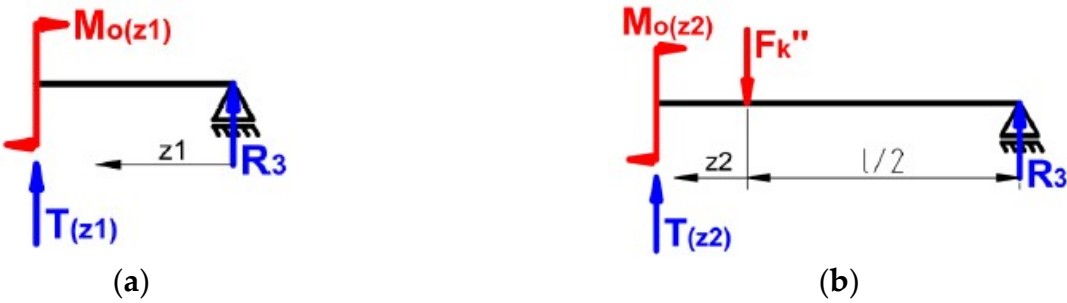

(**a**)            (**b**)

**Figure 22.** Application of the Schwedler's theorem: (**a**) a first cut of the upper guiding tube, (**b**) a second cut of the upper guiding tube.

Then, we proceed as follows:

- The first section:

$$z_1 \in \left\langle 0; \frac{l}{2} \right\rangle. \tag{66}$$

The bending moment $M_{O(z_1)}$ is:

$$M_{O(z_1)} = R_3 \cdot z_1 = \frac{F_k^{II} \cdot z_1}{2} \tag{67}$$

and the lateral force $T_{(z_1)}$ is calculated as:

$$T_{(z_1)} = -\frac{d\left(M_{O(z_1)}\right)}{d(z_1)} = -\frac{d\left(\frac{F_k^{II} \cdot z_1}{2}\right)}{d(z_1)} = -\frac{F_k^{II}}{2}. \tag{68}$$

- The second section:

$$z_2 \in \left\langle 0; \frac{l}{2} \right\rangle. \tag{69}$$

The bending moment $M_{O(z_2)}$ is:

$$M_{O(z_2)} = R_3 \cdot \left( \frac{l}{2} + z_2 \right) - F_k^{II} \cdot z_2 = \frac{F_k^{II} \cdot l}{4} - \frac{F_k^{II} \cdot z_2}{2} \tag{70}$$

and the lateral force $T_{(z_2)}$ is calculated as:

$$T_{(z_2)} = -\frac{d\left(M_{O(z_2)}\right)}{d(z_2)} = -\frac{d\left( \frac{F_k^{II} \cdot l}{4} - \frac{F_k^{II} \cdot z_2}{2} \right)}{d(z_2)} = -\frac{F_k^{II}}{2}. \tag{71}$$

The entire calculation is performed by means of the differential equation of the elastic curve as follows:

$$w = \int\limits_0^{z_1} \frac{M_{O(z_1)}}{E \cdot J_x} \cdot \frac{\partial M_{O(z_1)}}{\partial F_{kII}} \cdot dz_1 + \int\limits_0^{z_2} \frac{M_{O(z_2)}}{E \cdot J_x} \cdot \frac{\partial M_{O(z_2)}}{\partial F_{kII}} \cdot dz_2. \tag{72}$$

After integration and substituting corresponding parameters, the differential equation of the elastic curve is:

$$w = \frac{1}{E \cdot J_x} \cdot \left[ \int\limits_0^{\frac{l}{2}} \frac{F_{kII} \cdot z_1}{2} \cdot \frac{\partial \frac{F_{kII} \cdot z_1}{2}}{\partial F_{kII}} \cdot dz_1 + \int\limits_0^{\frac{l}{2}} \left( \frac{F_{kII} \cdot l}{4} - \frac{F_{kII} \cdot z_2}{2} \right) \cdot \frac{\partial \left( \frac{F_{kII} \cdot l}{4} - \frac{F_{kII} \cdot z_2}{2} \right)}{\partial F_{kII}} \cdot dz_2 \right]. \tag{73}$$

where $z_1$ and $z_2$ are sections of the analysed track and $F_{kII}$ is the force acting on the track in the analysed sections. The complete calculation is very extensive; therefore, it cannot be completely described.

Comparing results shown in Figures 23 and 24, we can see that both load cases are very similar even with the same values of the deflection. Therefore, we assume that the load transmission by this tube does not have the significant effect on the limit value of the platform deflection.

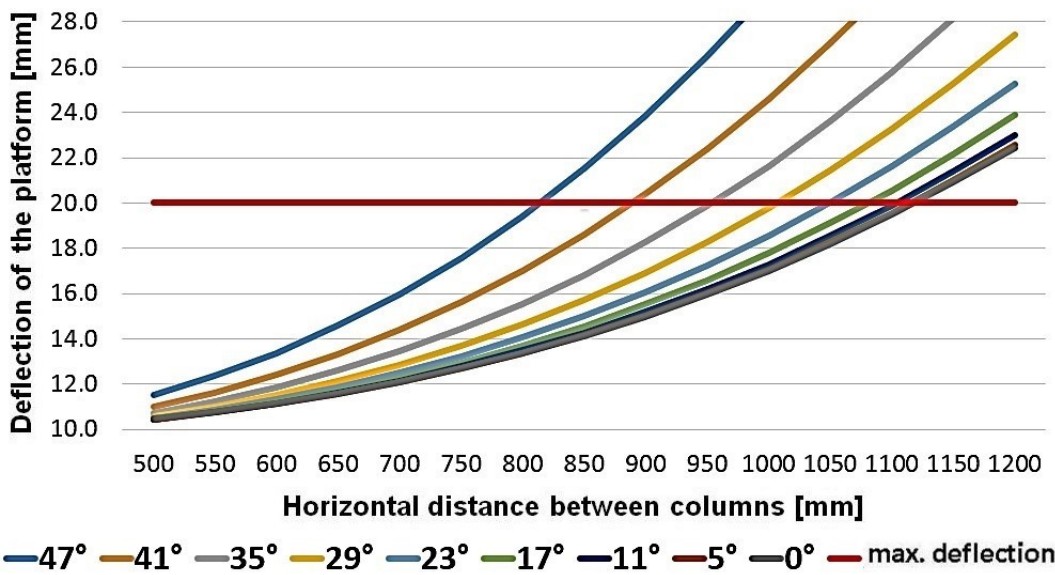

**Figure 23.** Dependence of the platform deflection on the horizontal distance between columns, when the load is transmitted by the upper tube. Results are shown for different track slope from 0° to 47°.

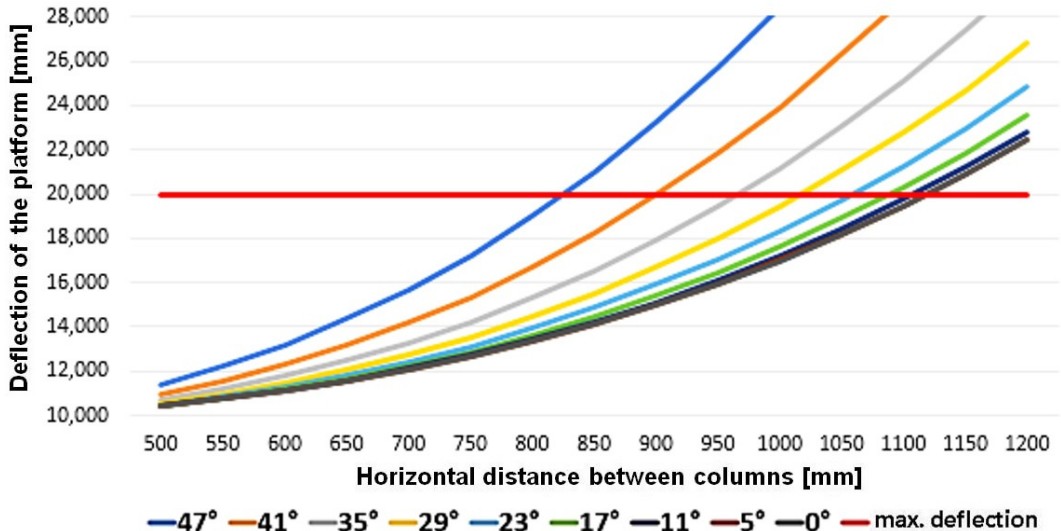

**Figure 24.** Dependence of the platform deflection on the horizontal distance between columns, when the load is transmitted by the bottom tube. Results are shown for different track slope from 0° to 47°.

The analysis of the load distribution state goes before the calculation of the guiding tube. The allowed overload of the platform with the weight of 375 kg is only in the location of the first column, i.e., in the location where the vector straight line of the gravitational force and the main central axis of the column are in the same plane. This plane is also perpendicular to the platform movement. Hence, it is not appropriate to analyse the tube in this location, because the tube is mounted just in this location. Suitably, the platform is imagined to move to the geometric centre of the selected computational track section, i.e., between two columns considering the load of 300 kg. In case of a greater load, the platform must not move; therefore, the situation when the platform with the load of 375 kg is in the vertical symmetry axis, cannot happen.

## 7. Conclusions

The article presents a partial solution to the conception of the design of a manipulator for handling with goods in particular difficult and limited operation conditions. It is considered to transport material between two destinations.

While the problem is extensive, the partial objective can be considered to have been met. The engineering design presented in this article fully complies with the requirements that will be imposed on it in operation. Analytical and numerical calculations implemented to the solution process completely support this claim.

The need of application of the manipulator in the production process of the company only demonstrates the necessity of mechanisation and automation technology and its invaluable importance in all areas.

The further solution of the device will include the design and analysis of the track and creation of a mathematical model for verifying the safety of the track in collaboration with the designed platform as well as drawing documentation, production and installation of the platform to the real operation.

**Author Contributions:** Conceptualization, M.B. and E.K.; methodology, J.D., M.B., M.S. and J.G.; software, J.D. and E.K.; validation, M.S., M.B. and J.G., formal analysis, J.D., M.B. and E.K.; investigation, M.B. and E.K.; J.D. and J.G.; writing—original draft preparation, J.D., M.B. and E.K.; writing—review and editing, M.S. and J.G.; visualization, M.B. and E.K.; supervision, M.S. and J.G.; funding acquisition, M.S. and J.G. All authors have read and agreed to the published version of the manuscript.

**Funding:** This work has been supported by grant agency grant agency KEGA project No. 001ŽU-4/2020.

**Conflicts of Interest:** The authors declare no conflict of interest.

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
