# Peer review of "Design of a Mechanical Part of an Automated Platform for Oblique Manipulation"

_applsci, doi:10.3390/app10238467_

Round 1

Reviewer 1 Report

In the introduction to the work entitled Design of A Mechanical Part of An Automated  Platform for Oblique Manipulation the authors pointed to the importance and wide application of modern warehouse.

A modern warehouse for a set of modern systems, transport, lifting, sorting and packing machines Which develop the set of transport lines, and The choice of the transport system in the warehouse is a decisive stage in its design. The correct selection of the system has an impact on technical, functional, economic and ergonomic aspects

In the second chapter, the authors point to the many factors that should be taken into account when choosing a system. The authors formulated them correctly.

The third chapter presents the structure of the handling machine consists of two separate but interconnected parts.The authors defined all the external loads acting on the device and the relations between them. They interpreted their analytical calculations correctly.

Chapter 4. The authors devoted to the selection and analysis of the most loaded element of the platform. They rightly noticed that it is the output shaft of the gearbox. They conducted the necessary strength analyzes for him.

In the next 5th chapter, they analyzed the platform deflection in accordance with the STN EN 81-40 standard.

In the summary, the authors indicated their plans for the structure they developed.

My assessment of the work is positive but I ask the authors to comment on my comments below.

153 table 2 unit of gravity [m] (should be [N],)

Figure 12 in the drawing is MkR, shouldn't there be MBR - described in the text as the frictional moment in the bearings?

Generally It has been shown that the manipulator fulfills its useful functions. Basic strength analyzes were performed for the most loaded parts and the
engine power was determined. Although analyzes of the influence of selected parameters on the functionality of the device were carried out, including:
the influence of the x value (position of the center of gravity)
on the critical values, or the angle of origin of the guide rail
on the engine load, it is possible to realize, however, that the scientific significance of the work was poorly emphasized, which is the novelty of the proposed solution (is such a solution not used yet?), do the authors see the possibility of optimizing dimensions to minimize the transferred load and power of the device?

Author Response

Dear reviewers,

Thank you for your reviews, which helped to improve the quality of our manuscript.

Reviewer 2 Report

This article presents a mechanism for mounting loads on an inclined plane. It is a new design of a "stair-climbing" mechanism without stairs. The mechanism is very classical and widely used in homes and industry. The specifications are clearly presented but the solution chosen is not sufficiently argued. Only a static study is described. The dynamics of the system for design and control is forgotten. Some elements of the calculations are presented. This is not research but engineering. The dynamic analysis with the law of motion to accelerate and stop the mechanism is missing. Rigidity analysis is incomplete.

Author Response

Dear reviewer,

thank you for your review, which helped to improve the quality of the manucript.

Reviewer 3 Report

The aim of the article seems to be the description of a design of the platform manipulator. There are few scientific elements in the work, but the problem is of an application nature - interesting enough to be published.

The main objection to the article lies in the fact that the article is unreadable. It looks a bit as if the authors wrote for themselves or only for people who know similar problems. Among other things, there is no description of the symbols and clear static diagrams. I will try to make some detailed comments below.

Page 3, Design of the accepted solution, lines 92-98. These sentences do absolutely nothing. In my opinion, they are obvious to everyone. Similarly, the sentence in lines 109-110: if it is designed badly, it can be a problem ...

Page 4, line 16. What does it mean "will mode"? It is a noun. Shouldn't be "will move" or something similar?

Page 4, Figure 5 alone is insufficient to understand the forces in the system. This point is very important in the article, and you have to try very hard to make the static diagram understandable. I believe that the layout should be drawn in two planes or in space.

Page 6, table 3. In the last row: the sum of Ran + Rbn does not agree with the values above (they are positive and negative).

Page 7, equation (11): the values of r and R should be explained here, the easiest way is to refer to Figure 8.

Page 8, line 196: where MBR [N∙m] is the friction moment in a bearing. OK, but where is this moment calculated? The value 69.7 appears in equation (23), but it is not known from where. This is generally a general comment: sometimes values appear "out of nowhere".

Page 9, equation (27). Another such example. Where did the omega value come from? Maybe I was looking wrong, but I didn't find it.

Page 12, Figure 12a: I have no idea where this moment diagram came from with the forces as in Figure 12a.

Page 13, lines 268-271: I do not understand these sentences and conclusions.

Page 14. Analysis of the platform deflection. It is difficult to understand this point, because the deflection is nowhere shown in the drawing, so it is difficult to say what the displacement is about. Of course, equation (41) is unclear because it is not known what the individual quantities mean.

Page 14-15, caption under Figure 16, but also line 294: "... when
the load is transmitted by the bottom tube". It is not well known why force should be transmitted only through the bottom tube or only through the upper tube.

There is not much in the conclusions (page 15), so it is difficult to comment on them. It is known that in the case of such work, the conclusions are (and will be) quite general.

Author Response

Dear reviewer,

thank you for your review, which helped to improve the quality of the manuscript.

Round 2

Reviewer 3 Report

The article has been significantly improved. Some aspects are explained which makes the article much easier to understand. The authors showed a lot of goodwill and work to improve the article. Even so, I have considerations that should be included (but don't require re-review):

  1. Figure 5 is of poor quality.
  2. Figures 16, 17, 18 and 19 have illegible descriptions (too small fonts).
  3. Figure 12. I certainly don't understand something, but I will share my doubts. In Figure 12b, force Ft acts downward, the moment diagram increases upward ... In Figure 12a, the force on the left (FR1) is moving up and the graph is rising up again. Besides, it is not known what the variable z1 measures (it looks as if the arrow should be in a different place).

Author Response

Dear reviewer,

thank you for your review.

Our responses are included in the attached file.
